# Rapid DNA Sequencing Technology Based on the Sanger Method for Bacterial Identification

**DOI:** 10.3390/s22062130

**Published:** 2022-03-09

**Authors:** Shunsuke Furutani, Nozomi Furutani, Yasuyuki Kawai, Akifumi Nakayama, Hidenori Nagai

**Affiliations:** 1Advanced Photonics and Biosensing Open Innovation Laboratory (Photo-BIO OIL), National Institute of Advanced Industrial Science and Technology (AIST), 2-1 Yamadaoka, Suita 565-0871, Osaka, Japan; shunsuke-furutani@aist.go.jp (S.F.); nozomi-furutani@aist.go.jp (N.F.); 2Department of Emergency and Critical Care Medicine, Nara Medical University, 840 Shijo, Kashihara 634-8522, Nara, Japan; k6k6k@naramed-u.ac.jp; 3Department of Medical Technology, School of Health Sciences, Gifu University of Medical Science, 795-1 Ichihiraga, Seki 501-3892, Gifu, Japan; anakayama@u-gifu-ms.ac.jp

**Keywords:** DNA sequence, DNA, PCR, cycle sequence, electrophoresis, Sanger method, rapid identification

## Abstract

Antimicrobial resistance, a global health concern, has been increasing due to inappropriate use of antibacterial agents. To facilitate early treatment of sepsis, rapid bacterial identification is imperative to determine appropriate antibacterial agent for better therapeutic outcomes. In this study, we developed a rapid PCR method, rapid cycle sequencing, and microchip electrophoresis, which are the three elemental technologies for DNA sequencing based on the Sanger sequencing method, for bacterial identification. We achieved PCR amplification within 13 min and cycle sequencing within 14 min using a rapid thermal cycle system applying microfluidic technology. Furthermore, DNA analysis was completed in 14 min by constructing an algorithm for analyzing and performing microchip electrophoresis. Thus, the three elemental Sanger-based DNA sequencing steps were accomplished within 41 min. Development of a rapid purification process subsequent to PCR and cycle sequence using a microchip would help realize the identification of causative bacterial agents within one hour, and facilitate early treatment of sepsis.

## 1. Introduction

The recent rise in antimicrobial-resistant bacteria due to improper use of antibacterial agents has become a global public health problem. Additionally, newly developed antibacterial agents may soon become unusable due to the development of new resistant strains [1]. Furthermore, pharmaceutical companies are not actively developing new antibacterial agents due to the difficulty in forecasting profits [2]. It is estimated that 10 million people will be infected annually with antimicrobial-resistant bacteria by 2050 [3]. Therefore, proper use of existing antibacterial agents is crucial for preventing the development of resistant strains. Action plans on antimicrobial resistance (AMR) have been set by various countries around the world, including Japan, and the use of appropriate antibacterial agents is being promoted. Additionally, technologies that realize appropriate use of antibacterial agents need to be developed [4].

Sepsis has a high mortality rate, and prompt administration of antibacterial agents is important. David et al. reported that administration of appropriate antibacterial agents (within one hour) significantly reduces mortality [5]. However, the current gold standard, blood culture method, takes several days to identify the causative bacteria [6]. Although matrix-assisted laser desorption/ionization time-of-flight mass spectrometry (MALDI-TOF MS) has recently attracted attention for rapid identification of bacteria, it can be used only for blood culture-positive samples due to the low limit of detection required to detect over 10,000 bacteria. Therefore, in both methods, broad spectrum antibacterial agents such as carbapenem are often used before the causative bacteria is identified, which increases the risk of antibacterial resistance developing. In particular, development of resistance to carbapenem-based antibacterial agents severely limits the option of therapeutic antibacterial agents making the treatment challenging.

Apart from the gold standard blood culture method, the causative bacterium can also be identified using gene-based analysis. Generally, bacterial species can be identified by analyzing the sequences of 16S ribosomal DNA (rDNA) generated using techniques such as Sanger sequencing, and then collated in databases. However, the conventional Sanger method is not widely used since it takes more than six hours to identify the bacterial species. Next-generation sequencing (NGS) technologies have recently been used to identify bacteria. Jangsup et al. identified the causative agent of bacterial meningitis by performing a polymerase chain reaction (PCR) and 16S rDNA analysis using nanopore sequencing [7]. Other researchers have also identified bacteria by conducting 16S rDNA sequencing using the nanopore sequencer [8,9,10]. However, although nanopore-based sequencing is fast (10–180 min), identification of bacteria within one hour, which is important for early treatment of sepsis, has not been achieved since 16S rDNA PCR amplification takes a long time. Rapid microchip electrophoresis using micro total analysis systems (μTAS) has been studied for many years [11,12,13]. Fredlake et al. achieved fast sequence analysis of 600 bases within 6.5 min by separating sequences at the 1-base level [14]. However, although microchip electrophoresis is fast, bacterial identification cannot be realized within the required time, as PCR and fluorescence labeling in the cycle sequencing reaction takes a long time.

We previously developed a rapid real-time PCR system, which takes at least 6.5 min [15]. In this study, by combining rapid PCR technology with microchip electrophoresis technology, we achieved rapid identification of bacteria using rapid DNA sequencing based on the Sanger method, which is more reliable for DNA sequence analysis. Figure 1 shows the clinical testing process and the whole experimental scheme for rapid DNA sequencing based on the Sanger method.

## 2. Materials and Methods

### 2.1. Rapid PCR

KOD One PCR Master Mix was purchased from TOYOBO CO., LTD (Japan). SpeedSTAR HS DNA polymerase and 16S rDNA primers for bacterial identification were purchased from TAKARA BIO INC (Kusatsu, Japan). The sequence of the 16S rDNA forward primer was 5′-GTT TGA TCC TGG CTC A-3′, while that of the reverse primer was 5′-TAC CAG GGT ATC TAA TCC-3′. The amplicon had an expected size of 793 bp. The SpeedSTAR HS DNA Polymerase PCR mixture consisted of 0.1 U μL^−1^ DNA polymerase with 1X Fast buffer I, 200 μM dNTPs, and 1.2 μM 16S rDNA primers, with concentrations optimized previously [15]. The KOD One PCR Master Mix consisted of 1X KOD One Master Mix and 1.2 μM 16S rDNA primers. Furthermore, 0.2X ROX Reference Dye (TAKARA BIO INC, Japan) and 120 μM Cyanin5-azide (SIGMA-ALDRICH CO. LLC, St. Louis, MO, USA) were added to the PCR mixtures since the rapid PCR system required a fluorescent dye. *Escherichia coli* was cultivated overnight at 37 °C in LB medium. The concentration of the *E. coli* was adjusted to 1 × 10^5^ cells μL^−1^ with nuclease-free water (Ambion, Austin, TX, USA).

For the rapid PCR amplification, the PCR mixture was incubated at 96 °C for 10 s as a hot start step to activate the DNA polymerase, followed by 45 cycles of 96 °C for 5 s and 55 °C for 4–18 s. The reactions were conducted on the GeneSoC system (KYORIN Pharmaceutical Co., Ltd., Tokyo, Japan). The PCR amplicon was purified using AMPure XP (Beckman Coulter, Brea, CA, USA), and 1% agarose gel electrophoresis was used to identify the shortest extension time required for amplification of the target product.

### 2.2. Rapid Cycle Sequencing for Fluorescent Labeling

BigDye Cycle Sequence v1.1 (Applied Biosystems, Waltham, MA, USA) was used for fluorescence labeling. BigDye terminators are labeled with the dRhodamine acceptor dyes: dR110 for guanine, dR6G for adenine, dTAMRA for thymine, and dROX for cytosine. The peak wavelengths of fluorescence intensities indicating guanine, adenine, thymine, and cytosine are approximately 540 nm, 570 nm, 595 nm, and 625 nm, respectively. The 16S rDNA forward primer was used at a final concentration of 480 nM. A total of 10 ng of the template DNA was used to amplify 16S rDNA. For the rapid cycle sequencing, the cycle sequencing reagent was incubated at 96 °C for 10 s as a hot start step to activate the polymerase, followed by 3–25 cycles of denaturation at 96 °C for 5 s, and annealing and extension at 55 °C for 20–120 s on the GeneSoC platform. After the reaction, the products were purified using CleanSEQ (Beckman Coulter, Brea, CA, USA), and analyzed on a capillary DNA sequencer (SeqStudio Genetic Analyzer, ThermoFisher, Waltham, MA, USA). The shortest time required for the reaction was identified based on the length of the sequence that could be analyzed and the fluorescence intensity of the longest fluorescent-labeled sequence. The minimum number of cycles required was confirmed based on the fluorescence intensity.

### 2.3. PCR and Cycle Sequencing Using Conventional Instruments

To evaluate the amplicons obtained using rapid PCR and rapid cycle sequencing, PCR and cycle sequencing were performed using conventional instruments (TaKaRa PCR Thermal Cycler Dice Gradient, TAKARA BIO INC, Kusatsu, Shiga, Japan). The PCR mixture consisted of 0.025 U μL^−1^ SpeedSTAR HS DNA Polymerase in 1X Fast buffer I, 200 μM dNTPs, and 0.5 μM 16S rDNA primers. *E. coli* was cultivated overnight at 37 °C in LB medium and the concentration adjusted to 1.0 × 10^5^ cells μL^−1^ with nuclease-free water. DNA controls for *Mycoplasma pneumoniae* and *Streptococcus pneumoniae* (Amplirun, Vircell, Granada, Spain) were also included. The concentration of each DNA control was adjusted to 1.0 × 10^4^ genome copies μL^−1^ with nuclease-free water. The PCR mixture was incubated at 96 °C for 10 s as a hot start step to activate the DNA polymerase, followed by 50 cycles of denaturation at 96 °C for 5 s, and extension at 55 °C for 30 s. PCR amplicons were purified using AMPure XP.

BigDye Cycle Sequence v1.1 was used for fluorescence labeling. The 16S rDNA forward primer was used at a final concentration of 160 nM. The cycle sequencing reagents were incubated at 96 °C for 60 s as a hot start step to activate the polymerase, followed by 25 cycles of 96 °C for 10 s, 50 °C for 5 s and 60 °C for 240 s. PCR products were subsequently purified using CleanSEQ.

### 2.4. Microchip Electrophoresis

DNA sequence analysis was performed by constructing an optical system (Figure 2). For detection, the microchip electrophoresis was followed by irradiation with an Ar laser (GLS3050, NEC, Tokyo, Japan) at 488 nm to excite the fluorescent dye through a 20X objective lens (CFI S Plan Fluor LED, Nikon, Tokyo, Japan) of an ECLIPSE Ti2-U microscope (Nikon, Japan). Fluorescence from the four emitted wavelengths were separated using dichroic mirrors, and measured using four photomultiplier tubes (PMTs) (Hamamatsu, Japan). The band pass filter 1 (BPF-1, FF01-482/35), the dichroic mirror 1 (DM-1, FF506-Di03), and the long pass filter (LPF, FF01-500/LP) were purchased from Nikon (Japan). The BPF-2 (FF01-540/15-25), BPF-3 (FF01-567/15-25), BPF-4 (FF01-590/20-25), BPF-5 (FF01-625/15-25), DM-2 (FF580-FDi01-25 × 36), DM-3 (FF555-Di03-25 × 36), and DM-4 (FF605-Di02-25 × 36) were purchased from Semrock (USA). The software for detection was developed inhouse using LabVIEW software, and graphed the signal obtained at a sampling rate of 5 Hz.

The microchip for electrophoresis and DNA sequencing was purchased from Micronit (Germany). This microchip has a cross-shaped introduction part with a flow path width of 50 μm. The separation length from the injection point to the detection point was set to 60 mm. Based on the study by Fredlake et al. [14], DNA sequencing was carried out in poly(N,N-dimethylacrylamide) (PDMA) solution containing 1X TTE buffer (49 mM Tris, 49 mM N-(Tris(hydroxymethyl)methyl)-3-aminopropanesulfonic acid, and 2 mM EDTA) with 7 M urea. The PDMA solution was composed of a mixture of 3% large molecular weight PDMA (Mw = 2.3 × 10^6^, NARD Institute Ltd., Amagasaki, Japan) and 2% small molecular weight PDMA (Mw = 3.4 × 10^5^, Polymer Source Inc., Montreal, QB, Canada). A custom-made high voltage sequencer (Hamamatsu, Japan) was used for sample introduction and separation. Figure 3 shows the positions of voltage application during sample introduction and separation in the microchip. The sample was placed in Reservoir 1, and PDMA solution was placed in Reservoirs 2 to 4. The sample was injected for 150 s at 400 V cm^−1^. Separation was carried out at 235 V cm^−1^ with 400 V cm^−1^ back-biasing applied to the sample and sample waste wells to eliminate sample leakage during the separation step.

### 2.5. Algorithm for Analyzing Microchip Electrophoresis

The analysis algorithm of the microchip electrophoresis result was developed with reference to the papers of Ewing et al. [16]. Fluorescence data from microchip electrophoresis were analyzed as base sequences using the following algorithm: First, to correct the background, each fluorescent value was subtracted from the minimum fluorescent value of 50 points before and after the measured value. Next, the large peak position indicating the start of the signal was recognized, and the fluorescent values before the signal were excluded. Then, each fluorescent value was normalized to the maximum fluorescent value of 75 points before and after the measured value. The normalized fluorescent values were noise-filtered according to the following equation:A_n_′ = (A_n−1_ + 2 × A_n_ + A_n+1_)/4(1)
where A_n_ and A_n_′ represent the nth fluorescent value and the nth filtered fluorescent value, respectively. Since the four fluorescent color dyes affect the long wavelength side when each dye emits fluorescence, the fluorescence values for adenine, thymine, and cytosine were corrected based on the fluorescent value for guanine, using the following equation:F_A_′ = F_A_ − 0.489 × F_G_(2)
F_T_′ = F_T_ − 0.273 × F_G_(3)
F_C_′ = F_C_ − 0.122 × F_G_(4)
where F_G_, F_A_, F_T_, and F_C_ represent the fluorescent values for guanine, adenine, thymine, and cytosine, respectively, and F_A_′, F_T_′ and F_C_′ stand for F_A_, F_T_ and F_C_ corrected based on F_G_, respectively. Furthermore, F_A_′, F_T_′ and F_C_′ were normalized to their respective overall values, and these values were set as normalized F_A_′, normalized F_T_′ and normalized F_C_′, respectively. The fluorescence values for thymine and cytosine were corrected based on the fluorescent value for adenine using the following equation:F_T_″ = Normalized F_T_′ − 0.406 × Normalized F_A_′(5)
F_C_″ = Normalized F_C_′ − 0.200 × Normalized F_A_′(6)
where F_T_″ and F_C_″ represent normalized F_T_′ and normalized F_C_′ corrected based on normalized F_A_′, respectively. Furthermore, F_T_″ and F_C_″ were normalized to their respective overall values, and these values set as normalized F_T_″ and normalized F_C_″, respectively. Finally, the fluorescence values for cytosine were corrected based on the fluorescent value for thymine according to the following equation:F_C_‴ = Normalized F_C_″ − 0.563 × Normalized F_T_″(7)
where F_C_‴ represents normalized F_C_″ corrected by normalized F_T_″. Furthermore, F_C_‴ was normalized to its respective overall value, and the values were set as normalized F_C_‴. Since F_G_, normalized F_A_′, normalized F_T_″, and normalized F_C_‴ have different rates of electrophoresis separation for each fluorescent dye, the mobility was corrected by shifting the signals of F_G_, normalized F_T_″, and normalized F_C_‴ back by 0.6 s, 0.4 s and 0.2 s, respectively. Finally, nucleotide sequence analysis was performed using the peak position 0.5X or more of the maximum fluorescence value at each time and each base.

## 3. Results & Discussion

### 3.1. Rapid PCR

We examined the rapid PCR conditions for 16S rDNA amplification using GeneSoC, which is a rapid real-time PCR system. In this system, rapid thermal cycling is realized by reciprocating the PCR reagents on two heaters [15]. GeneSoC uses a fluorescent probe for gene quantification, and a fluorescent signal is required to control the reciprocating liquid feed. In this experiment, amplification of only the target DNA is required for sequencing, since there is no need to quantify the initial concentration or confirm amplification using a fluorescent probe. Rapid PCR conditions were optimized by adding a fluorescent dye to the PCR reagent in GeneSoC. In our previous studies, SpeedSTAR DNA HS Polymerase was used as the optimal enzyme for rapid fluorescence-based PCR amplification using the TaqMan probe method [15,17,18]. In the present study, we additionally confirmed rapid PCR amplification using KOD One polymerase, which lacks 5′ nuclease activity. Figure 4 shows the results of 1% agarose gel electrophoresis following 45 cycles of PCR amplification with 5 s denaturation and 4–18 s extension times; 793 bp 16S rDNA amplicons were confirmed in samples subjected to extension of 10 s or more using SpeedSTAR DNA HS polymerase, and 8 s or more using KOD One polymerase. In addition, smear-like amplicons longer than 793 bp were confirmed in samples subjected to extension of 12 s or longer using both SpeedSTAR DNA HS and KOD one polymerases. Since these smears can generate noise signals during sequencing, we chose 8 s as the optimal annealing and extension time for rapid PCR amplification using KOD one polymerase. Under these conditions, amplification of 16S rDNA required for bacterial identification could be achieved in 45 cycles and within 13 min. As the sensitivity, the limit of detection (LOD) by PCR for 16S rDNA has been reported to be 1–10 cfu μL^−1^ [19]. In addition, it has been reported as 10 copies per reaction as a LOD using clinical specimens in rapid PCR using GeneSoC [20]. Therefore, the result of this study is also expected to be around 10 copies per reaction as LOD.

### 3.2. Rapid Cycle Sequencing for Fluorescence Labeling

Reaction conditions for rapid cycle sequencing were optimized on the GeneSoC platform. The fluorescent-labeled products were analyzed using a capillary DNA sequencer, and the minimum duration required to detect the amplification product was evaluated. Figure 5a shows the ratio of fluorescence values at the end portion of the amplification product versus the maximum fluorescence intensity. Appendix A shows raw data from a capillary DNA sequencer. The fluorescence intensity generated by the amplified product did not increase significantly when the reaction duration was 20 s or 30 s. These results indicate that the enzymatic elongation reaction and fluorescence labeling during cycle sequencing were inadequate. Figure 5b shows the number of base sequences that were analyzed at each reaction duration. When the reaction duration was 30 s or less, the number of base sequences analyzed varied widely, and analysis could not be performed to the end. However, when the reaction duration was 40 s or more, the variation in the number of analyzed bases was small, but the analysis could be performed to the end. These results confirm that the cycle sequencing reaction duration can be shortened to 40 s.

The number of cycles required for fluorescent labeling was also examined, as it is not sufficient to only obtain the fluorescence intensity required for DNA sequence analysis. Figure 6 shows the correlation between fluorescence intensity and the number of cycles for fluorescent labeling. Fluorescence intensity increased in proportion to the number of cycles, for up to 25 cycles. However, when performing cycle sequencing using the general method, the fluorescence intensity reached 1074 ± 223 when the reaction was carried out for 25 cycles. These results confirmed that approximately 15 cycles were sufficient for rapid cycle sequencing on the GeneSoC platform. As the primer concentration used was three-fold higher that used in the rapid PCR reaction, we considered that the reaction did not reach saturation, and the fluorescence intensity increased up to the 25th cycle. When the cycle sequencing reaction duration was set at 40 s and carried out for 15 cycles, rapid cycle sequencing was achieved within 14 min.

### 3.3. Microchip Electrophoresis

Microchip electrophoresis and analysis algorithms developed in this study were evaluated using samples subjected to PCR and cycle sequencing. Figure 7 shows the electrophoresis data for *E. coli,* and an enlarged view of the analysis results generated using the developed algorithm. Fluorescence intensity peaks corresponding to dR110 (guanine), dR6G (adenine), dTAMRA (thymine), and dROX (cytosine) were confirmed, and 550 bases identified using this algorithm. Appendix A shows the results of an alignment of a blast search carried out on the National Center for Biotechnology Information (NCBI) website. We confirmed that the sequence belonged to *E. coli* 16S rDNA, with a 96% identity (Appendix A). Thus, DNA sequence analysis was accomplished using microchip electrophoresis and the developed analysis algorithm. The number of bases sequenced using capillary electrophoresis was less than 750 (Appendix A). The main reason for this is that the separation length in a microchip was 60 mm, which is approximately 1/5 that of capillary electrophoresis, thus sequences after the 550th base could not be accurately separated. However, the duration required for sample introduction and separation was less than 14 min, approximately 1/8 that of capillary electrophoresis. In microchip electrophoresis, a plug length of about 50 μm can be realized, as the sample is cut out in the shape of a plug at the cross section in the microchip. Meanwhile, in capillary electrophoresis, it is necessary to introduce the sample coaxially with the separation direction, thus a plug length of approximately several millimeters is required to introduce the sample with good reproducibility. In electrophoresis, the shorter the plug length, the shorter the separation duration. Therefore, microchip electrophoresis enables the shorter separation duration than capillary electrophoresis. Furthermore, *M. pneumoniae* and *S. pneumoniae* DNA sequences could be determined using microchip electrophoresis (Appendix A). The results show that these bacteria can be accurately identified using the developed analysis algorithm following microchip electrophoresis, with identities of 92% and 90% for *M. pneumoniae* and *S. pneumoniae,* respectively.

After confirming that bacterial identification was possible using the developed algorithm, we analyzed samples subjected to rapid PCR and rapid cycle sequencing to achieve rapid bacterial identification. A BLAST search on the NBCI database identified the sequence of the DNA sample subjected to 15 cycles of rapid cycle sequencing as belonging to *E. coli* 16S rDNA, with 90% identity. In addition, DNA sequence analysis is possible if sufficient fluorescence intensity is obtained, thus we tested whether this could be detected by subjecting a sample to 15 or fewer cycles to achieve rapid bacterial identification. Unfortunately, DNA analysis using microchip electrophoresis could not be determined using 10 or less cycles due to low fluorescence intensity. Our results showed that 15 cycles of rapid cycle sequencing were achieved within 14 min.

## 4. Conclusions

To achieve rapid DNA sequencing using the Sanger method, we optimized rapid PCR, rapid cycle sequencing, and microchip electrophoresis. Rapid PCR amplification of 16S rDNA for bacterial identification could be achieved in 45 cycles within 13 min. Rapid cycle sequencing generated samples with sufficient fluorescence intensity for sequence analysis in 15 cycles within 14 min. DNA sequence analysis using microchip electrophoresis was achieved within 14 min. *E. coli*, *M. pneumoniae*, and *S. pneumoniae* were identified in this study. Thus, the three elemental Sanger-based DNA sequencing steps were completed within 41 min. Table 1 summarizes a comparison of bacterial identification by different sequencing methods. Future development of rapid purification techniques following PCR and cycle sequencing using a microchip should facilitate the identification of causative bacterial agents within one hour, which is important for the early treatment of sepsis.

## Figures and Tables

**Figure 1 sensors-22-02130-f001:**
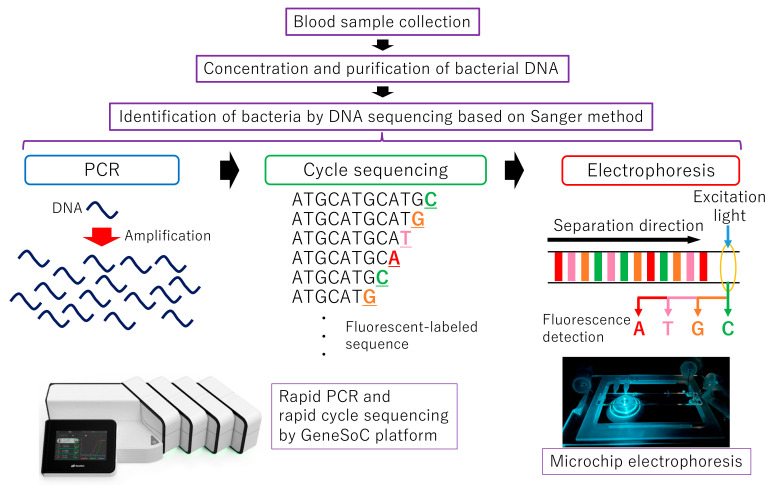
The clinical testing process and the whole experimental scheme for rapid DNA sequencing based on the Sanger method.

**Figure 2 sensors-22-02130-f002:**
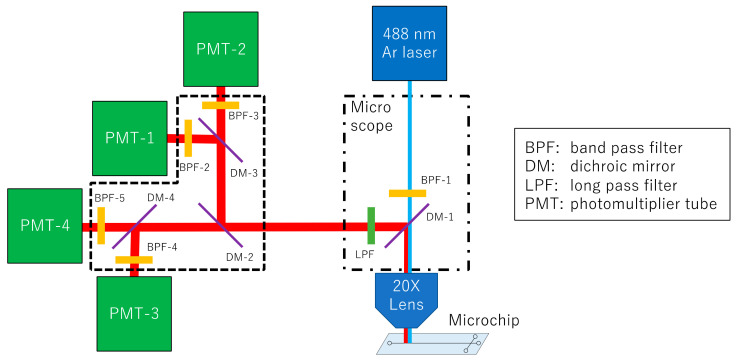
Schematic diagram of the optical system for DNA sequencing. The system consists of a 488 nm laser, a microscope with a 20X objective lens, and four PMTs. Fluorescence is split into four wavelengths using three dichroic mirrors, and detected using four PMTs through each band pass filter.

**Figure 3 sensors-22-02130-f003:**
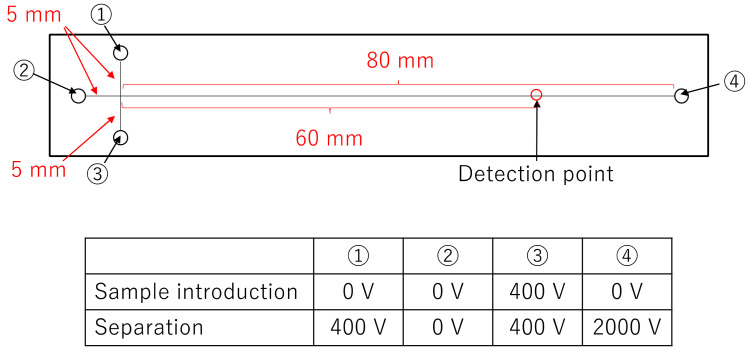
The sizes of microchannel in the microchip and the positions of voltage application during sample introduction and separation in the microchip.

**Figure 4 sensors-22-02130-f004:**
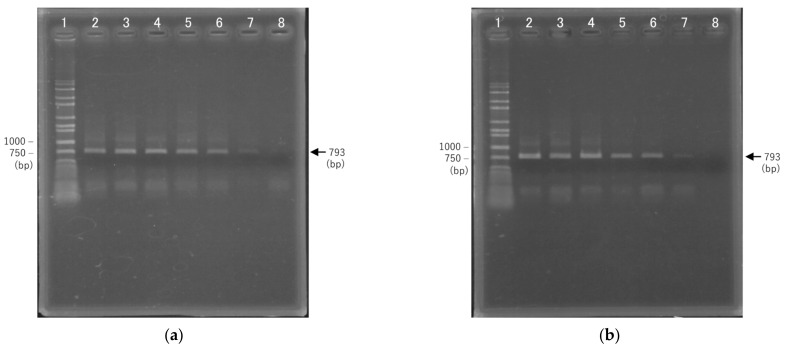
Results of 1% agarose gel electrophoresis of amplification products following 45 cycles of PCR using two DNA polymerases. (**a**) SpeedSTAR DNA HS Polymerase; 1: wide range DNA ladder, 2: 18 s, 3: 16 s, 4: 14 s, 5: 12 s, 6: 10 s, 7: 8 s, 8: 6 s. 2–8 indicate the annealing and extension durations. (**b**) KOD One polymerase; 1: wide range DNA ladder, 2: 16 s, 3: 14 s, 4: 12 s, 5: 10 s, 6: 8 s, 7: 6 s, 8: 4 s. 2–8 indicate the annealing and extension durations.

**Figure 5 sensors-22-02130-f005:**
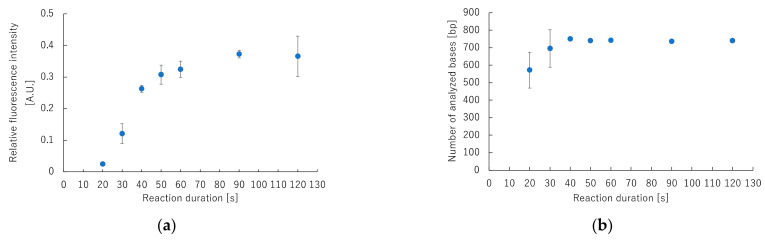
Correlation between fluorescence intensity and number of bases that can be analyzed for each reaction duration of cycle sequencing using capillary DNA sequencer. (**a**) Correlation between relative fluorescence intensity and reaction duration. (**b**) Correlation between the number of analyzed bases and reaction duration.

**Figure 6 sensors-22-02130-f006:**
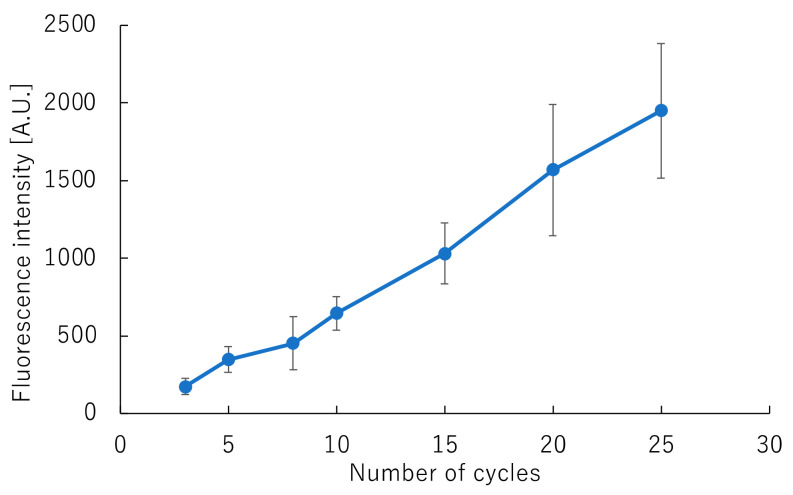
Correlation between fluorescence intensity and the number of fluorescence-labeling cycles.

**Figure 7 sensors-22-02130-f007:**
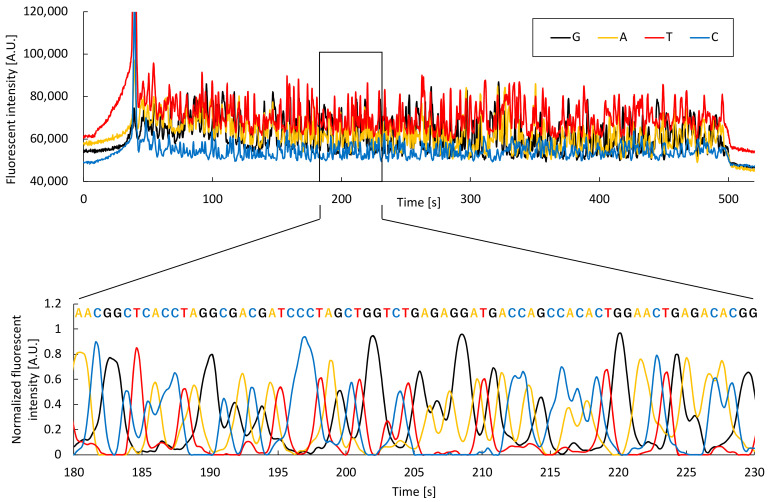
Electrophoresis data for *E. coli* and an enlarged view of the analysis based on the developed algorithm.

**Table 1 sensors-22-02130-t001:** The comparison of bacterial identification by different sequencing methods.

DNA Sequencing Method	Required Process	Analysis Time	Read Length	Reference
Traditional Sanger method	PCR	9 h or less	700–900 bp	[21]
	Cycle sequencing			
	Capillary electrophoresis			
Nanopore method	PCR	4.2–7.5 h	10,000 bp	[7]
	Library preparation			
	Nanopore sequencing			
Rapid Sanger method	Rapid PCR	41 min	550 bp	This study
	Rapid cycle sequencing			
	Microchip electrophoresis			

## Data Availability

The data presented in this study are available from the corresponding author upon reasonable request.

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
