# Peer review of "Rapid DNA Sequencing Technology Based on the Sanger Method for Bacterial Identification"

_sensors, 2022, doi:10.3390/s22062130_

Round 1
Reviewer 1 Report
General comment:
The study presented a rapid DNA sequencing based on the Sanger method, by optimizing rapid PCR, rapid cycle sequencing, and microchip electrophoresis. The optimized process can be completed within an hour. And E. coli, M. pneumoniae, and S. pneumoniae were identified with identities all greater than 90%. Overall, the experiment of this work was well organized and the results should be interest of the readership of Sensors. However, some issues should be addressed before the consideration of publication.
Specific comments:
1 Some references have errors in the writing of format, such as P9 L332.
2 Please supplement a scheme image to explain the whole experiment.
3 It is better to present the comparison of different sequencing methods in a tabular form, giving reader a sense of clarity at a glance.
4 The “Algorithm for analyzing microchip electrophoresis”passages lacks relevant literature and data references.
5 If the method is applied to clinical testing, please describe the entire testing process, focusing on the connection between laboratory testing and clinical testing,
Author Response
We are very grateful to the Editor and the referees for their comments on the revised version of our manuscript. Herein, we have provided a point-by-point response to the referee’s comments, and hope that the explanations and revisions are satisfactory.
Reviewer 1
General comment:
The study presented a rapid DNA sequencing based on the Sanger method, by optimizing rapid PCR, rapid cycle sequencing, and microchip electrophoresis. The optimized process can be completed within an hour. And E. coli, M. pneumoniae, and S. pneumoniae were identified with identities all greater than 90%. Overall, the experiment of this work was well organized and the results should be interest of the readership of Sensors. However, some issues should be addressed before the consideration of publication.
Specific comments:
1. Some references have errors in the writing of format, such as P9 L332.
We are thankful to the reviewer for this comment. We revised error in the writing of format in P11 L375.
2. Please supplement a scheme image to explain the whole experiment.
In accordance with the reviewer’s helpful input, P2 Figure 1 and P2 L73 - 74 “Figure 1 shows … on Sanger method.” were added in introduction.
3. It is better to present the comparison of different sequencing methods in a tabular form, giving reader a sense of clarity at a glance.
In accordance with the reviewer’s comment, P10 Table 1 and P9 L331 – 332 “Table 1 summarizes … different sequencing methods.” were added in conclusions.
4. The “Algorithm for analyzing microchip electrophoresis” passages lacks relevant literature and data references.
We are thankful to the reviewers for this comment. P5 L175 -176 “The analysis algorithm … Ewing et al [16].” and reference 16 were added.
5. If the method is applied to clinical testing, please describe the entire testing process, focusing on the connection between laboratory testing and clinical testing.
We are thankful to the reviewers for this important comment. Along with the revision for comment 2, P2 Figure 1 and P2 L73 - 74 “Figure 1 shows … on Sanger method.” were added in introduction.
Reviewer 2 Report
The Nagai group previously developed a rapid real-time PCR system called GeneSoC. In this study, they expanded this platform to identify bacterial species by including rapid cycle sequencing and microchip electrophoresis. Identification of bacteria in patients is important for prescribing the correct antibacterial agents, but the blood culture method, which is currently the standard, takes several days. The authors tested a new method which takes just 41 minutes. The manuscript presents an important methodological improvement and therefore is worth publication.
However, sequencing using microchip electrophoresis would be difficult for others to reproduce as the protocol described in this manuscript lacks essential details. I recommend the authors to include details about the custom-made high-voltage sequencer and the placement of electrodes for voltage application. Explanation of the difference between capillary and microchip electrophoresis methods would also make the paper more interesting to a general readership.
Author Response
We are thankful to the reviewer’s helpful comment. P5 Figure 3 and P4 L161 – 164 “Figure 3 shows … reservoirs 2 to 4.” were added to explain the details. Furthermore, P8 L295 -302 “In microchip electrophoresis, … than capillary electrophoresis.” was added to explain the difference between capillary and microchip electrophoresis methods.
Reviewer 3 Report
In this work, authors developed a rapid DNA sequencing technology based on Sanger method for bacterial identification, including rapid PCR method, rapid cycle sequencing, and microchip electrophoresis. The topic is significant and the result is satisfactory. However, I’d like to know the accuracy and sensitivity of the method you developed.
Author Response
We are thankful to the reviewer’s helpful comment. As for the accuracy, the identities for bacteria were already described in our manuscript. As for the sensitivity, P6 L228 – L232 “As the sensitivity, … reaction as LOD.” and references 19 and 20 were added.
Round 2
Reviewer 1 Report
This version addresses all my doubts and I agree to accept the manuscript.